# Effects of Synthetic Toll-Like Receptor 9 Ligand Molecules on Pulpal Immunomodulatory Response and Repair after Injuries

**DOI:** 10.3390/biom14080931

**Published:** 2024-08-01

**Authors:** Angela Quispe-Salcedo, Tomohiko Yamazaki, Hayato Ohshima

**Affiliations:** 1Division of Anatomy and Cell Biology of the Hard Tissue, Niigata University Graduate School of Medical and Dental Science, Niigata 951-8514, Japan; aquispesa@dent.niigata-u.ac.jp; 2Research Center for Macromolecules and Biomaterials, National Institute of Material Sciences (NIMS), Tsukuba 305-0047, Japan; yamazaki.tomohiko@nims.go.jp

**Keywords:** dental pulp, CpG oligonucleotide, Toll-like receptor 9, mice, odontoblasts, tooth replantation, tooth injuries

## Abstract

Synthetic oligodeoxynucleotides (ODNs) containing unmethylated cytosine–phosphate–guanine (CpG) motifs (CpG-ODNs) are ligand molecules for Toll-like receptor 9 (TLR9), which is expressed by odontoblasts in vitro and dental pulp cells. This study determined the effects of CpG-ODNs on pulpal immunomodulatory response and repair following injury. Briefly, the upper right first molars of three-week-old mice were extracted, immersed in Type A (D35) or B (K3) CpG-ODN solutions (0.1 or 0.8 mM) for 30 min, and then replanted. Pulpal healing and immunomodulatory activity were assessed by hematoxylin–eosin and AZAN staining, as well as immunohistochemistry. One week following the operation, inflammatory reactions occurred in all of the experimental groups; however, re-revascularization and newly formed hard tissue deposition were observed in the pulp chamber of all groups at week 2. A positive trend in the expression of immune cell markers was observed toward the CpG-ODN groups at 0.1 mM. Our data suggest that synthetic CpG-ODN solutions at low concentrations may evoke a long-lasting macrophage–TLR9-mediated pro-inflammatory, rather than anti-inflammatory, response in the dental pulp to modulate the repair process and hard tissue formation. Further studies are needed to determine the effects of current immunomodulatory agents in vitro and in vivo and develop treatment strategies for dental tissue regeneration.

## 1. Introduction

Since their discovery in the early 90s, synthetic oligodeoxynucleotides (ODNs) containing unmethylated cytosine–phosphate–guanine (CpG) motifs (CpG-ODNs) have been evaluated because of their high immunostimulatory activity. Comparable to bacterial DNA, synthetic CpG-ODNs trigger an immunostimulatory effect that leads to the maturation, differentiation, and proliferation of immune cells, such as B- and T lymphocytes, natural killer (NK) cells, monocytes, macrophages, and dendritic cells (DCs) [1,2]. Moreover, the recognition of CpG motifs is associated with Toll-like receptor (TLR) signaling. Early studies demonstrated that macrophages from TLR9 knockout mice did not respond to CpG DNA but to ligands for TLR2 and TLR4. Moreover, TLR9 expression in immune cells correlated with responsiveness to CpG DNA, inducing an innate immune response that triggers alterations in the cellular reduction–oxidation balance and the induction of cell signaling pathways, including the mitogen-activated protein kinases and necrosis factor kappa beta (NFκB) [3,4,5]. Regarding the structural and functional differences between both types, Type A CpG-ODNs contain a central palindromic CpG motif and form higher-order structures. They are primarily used for their strong interferon response, targeting plasmacytoid dendritic cells (pDCs) and eliciting systemic immunity. Type A CpG-ODNs also induce small amounts of inflammatory cytokines, such as IL-6. In contrast, Type B CpG-ODNs form linear structures and induce IL-6 secretion from TLR9-expressing cells, mainly B- and macrophage cells in mice. This results in the activation of B cells, promoting their proliferation and differentiation [1,2]. Current therapeutic applications of CpG-ODNs include their role as activators of innate immune defense against infections and potent agents for immunotherapy against cancer, allergies, asthma, and vaccine adjuvants [6,7]. Despite multiple applications from basic research to clinical medicine, their potential use in the dental field has not yet been evaluated.

Dental trauma, such as luxation injuries or avulsion, and replantation directly affect the dental pulp, periodontium, and surrounding bone, resulting in total severance of the neurovascular supply [8]. The outcome of pulpal severance will be either total pulpal re-vascularization or the development of partial or total pulp necrosis, which is dependent on the race between cellular ingrowth and bacterial invasion [8,9]. Following injury, an innate immune response occurs in the pulpal tissue, in which vascular alterations and inflammatory cell infiltration are activated to eliminate the irritating molecules [9]. Odontoblasts are located at the outermost cell layer of the dental pulp and are considered the first line of defense, as they are involved in the innate and adaptive immunity of the dental pulp against invading bacteria and noxious stimuli from the microflora of the oral cavity [10,11]. For example, when bacteria and their products invade deeply into dentinal tubules, odontoblasts are the first pulpal cells encountered by these dentin-invading microorganisms. They sense pathogen-associated molecular patterns (PAMPs) shared by microorganisms through specialized pattern recognition receptors (PRPs) at the dentin–pulp interface [12,13,14]. In vitro and in vivo studies have demonstrated that odontoblasts from humans and rodents express several PRRs, including members of the TLR family from 1–10 and nucleotide oligomerization binding domains. Interestingly, differentiated odontoblasts from mice, rats, and humans constitutively express TLR9 genes in vitro [11,12,15,16,17,18,19], which recognize CpG motifs in bacterial DNA and synthetic CpG-ODNs to elicit a tissular innate immune response. Exposure to CpG DNA induces a strong pro-inflammatory response in cultured mouse odontoblasts through the TLR9/MyD88/NF-κB signaling pathway. Similarly, exposure to CpG-ODNs induces a significant increase in interleukin (IL)-8 and matrix metalloproteinase (MMP)-13 expression through the TLR9, MyD88, NF-κB, and ERK pathways in odontoblast-like cells in mice [15,18,19]. Although informative, the available in vitro data on the immunostimulatory effects of CpG-ODNs in odontoblast-like cells should be analyzed in living organisms to determine the biological response to synthetic CpG-ODNs in pulpal tissue, particularly following injury.

We established a tooth replantation model using mice [20,21,22,23] to assess the chronological changes and cellular events that occur in dental pulp following injury. This model mimics the clinical situation of avulsion, in which a tooth is completely displaced from the alveolar socket, thus making it a reliable experimental design to evaluate the effects of experimental compounds on the pulpal tissue of extracted teeth following replantation. In this study, we analyzed for the first time the effects of synthetic CpG-ODNs, TLR 9 ligand molecules, on the pulpal immunomodulatory response and repair processes after injury using a mouse model for intentionally delayed tooth replantation.

## 2. Materials and Methods

### 2.1. In Vivo Experimental Procedures

#### 2.1.1. Animals

All animal experiments were conducted following the ARRIVE guidelines [24], and the protocol was reviewed by the Institutional Animal Care and Use Committee and approved by the President of Niigata University (approval #: SA00298 and SA01017).

Ninety-two male Crlj:CD1 Institute of Cancer Research (ICR) mice (three weeks old) were obtained from Charles River Laboratories Japan (Yokohama, Japan). The animals were divided into six experimental groups as follows: Distilled water (DW), Hank’s Balanced Salt Solution (HBSS, Fujifilm Wako Pure Chemical Industries, Ltd., Osaka, Japan), and CpG-ODN A and B groups at 0.1 mM and 0.8 mM concentrations (see Appendix A). The characteristics of the synthetic CpG-ODN solutions were as follows:

CpG-ODN A: Class A CpG oligonucleotide (human and mouse), endotoxin-free D35, partially phosphorothioated backbone, and the DNA sequence 5′-GGT GCA TCG ATG CAG GGG GG-3 (Catalog #: 65001; GeneDesign Inc., Osaka, Japan) tested at final concentrations of 0.63 mg/mL (0.1 mM) and 5 mg/mL (0.8 mM) in DW.

CpG-ODN B: Class B CpG oligonucleotide (human and mouse), endotoxin-free K3, phosphorothioated backbone, and the DNA sequence 5′-ATC GAC TCT CGA GCG TTC TC-3′, (Catalog #: 65003; GeneDesign) tested at final concentrations of 0.63 mg/mL (0.1 mM) and 5 mg/mL (0.8 mM) in DW.

#### 2.1.2. Tooth Replantation

The intentionally delayed tooth replantation injury model was performed, as described in our previous studies [20,21]. Briefly, the upper left and right first molars of each animal (one mouse, *n* = two teeth) were consecutively extracted under deep anesthesia following an intraperitoneal injection of a mixed solution (0.05–0.1 mL/10 g) comprising 1.875 mL of Domitor^®^ (Nippon Zenyaku Kogyo Co., Ltd., Koriyama, Japan), 2 mL of midazolam (Sandoz KK, Tokyo, Japan), 2.5 mL of Vetorphale^®^ (Meiji Seika Pharma Co., Ltd., Tokyo, Japan), and 18.625 mL of physiological saline using a pair of modified dental forceps. The extracted teeth were immersed for 30 min in the experimental solutions and then repositioned in their original sockets without further treatment, such as the fixation of teeth or relief of occlusion.

### 2.2. Tissue Preparation

Samples were collected from groups of 5–11 animals on weeks 1 and 2 following the operation. At each observation point, the mice were transcardially perfused with physiological saline, followed by 4% paraformaldehyde (PFA; Sigma-Aldrich, St. Louis, MO, USA) in 0.1 M of phosphate buffer (pH 7.4) under deep anesthesia with an intraperitoneal injection of a mixed solution of Domitor^®^, midazolam, Vetorphale^®^, and physiological saline. The maxillae were removed en bloc and immersed in 4% PFA for another 24 h at 4 °C. After decalcification in Morse’s solution (10% sodium citrate and 22.5% formic acid) for 4–5 days at 4 °C, the samples were dehydrated through a series of ethanol grades, embedded in paraffin, and sagittally cut at a thickness of 4 µm. The sections were then processed for hematoxylin–eosin (H&E) staining, AZAN staining, and immunohistochemistry.

### 2.3. Immunohistochemical Procedures

Sections were processed using the Envision + Horseradish Peroxidase System (Dako Japan, Tokyo, Japan; catalog #: K5027) and a mouse anti-nestin monoclonal antibody diluted to 1:200 (Millipore, Temecula, CA, USA; catalog #: MAB353) and a rabbit polyclonal anti-mannose receptor (CD206) antibody diluted to 1:1000 (Abcam, Cambridge, MA, USA; catalog #: ab64693). The avidin–biotin–peroxidase complex (Vectastain ABC Kit, Vector Laboratories, Burlingame, CA, USA) method was performed using a mouse monoclonal anti-Ki67 antibody diluted to 1:100 (Dako Japan; catalog #: M7249), a rat monoclonal anti-F4/80 antibody diluted to 1:250 (Novus Biologicals USA, Centennial, CO, USA; catalog #: NB600-404), and a mouse monoclonal anti-TLR9 antibody diluted to 1:200 (Abcam; catalog #: ab12121), along with a biotinylated anti-mouse IgG (H+L) diluted to 1:100 (Vector Laboratories; catalog #: BA-2001) for Ki-67 and TLR9, and a biotinylated anti-rat IgG (H+L) diluted to 1:100 (Vector Laboratories; catalog #: BA-4000) for F4/80. To visualize the sections, 0.05 M of Tris-HCl buffer (pH 7.6) containing 0.04% 3-3′-diaminobenzidine tetrahydrochloride and 30% H_2_O_2_ was used. The sections were counterstained with hematoxylin.

### 2.4. Cell Counting and Statistical Analysis 

Data analyses from H&E staining and immunohistochemistry at weeks 1 and 2 were analyzed using Image J software (Image J 1.53k, National Institutes of Health, Bethesda, MD, USA). The percentage of nestin-positive perimeters was calculated relative to the total perimeter of the pulp–dentin border. Similarly, the rate of newly formed hard tissue areas was calculated relative to the total area of the dental pulp. The percentage of cell density for Ki-67- and CD206-positive cells in the pulpal area of each specimen was obtained using a mine-squared grid of 2600 µm^2^ (total counting area: 23,400 µm^2^). For the analysis of F4/80 and TLR9 immunostaining, the percentage of the area fraction or the stained area relative to the total area of the pulp chamber was quantitated by adjusting the threshold using the plug-in Color Deconvolution 2 for Image J. All data are presented as the mean and standard deviation for each group. Statistical analysis was performed using IBM^®^ SPSS^®^ software (Ver 21, IBM, Tokyo, Japan). Data normality was analyzed with the Shapiro–Wilk test. For comparisons between groups, Bonferroni’s test for multiple comparisons was used after the confirmation of data normality and homogeneity of variance. The samples showing no normal distribution were compared by the Kruskal–Wallis test for more than three groups or the Mann–Whitney U test for two groups.

## 3. Results

### 3.1. Histological Evaluation of the Dental Pulp Healing Process by H&E and AZAN Staining and Nestin Immunohistochemistry

H&E and AZAN staining were performed to evaluate the morphological changes in the pulpal tissue and the presence of collagen-related matrices in the dental pulp, respectively. Furthermore, nestin was used as a marker to identify surviving odontoblasts and/or newly differentiated odontoblast-like cells. The progression of pulpal healing was determined by the percentage of the nestin-positive perimeter at the pulp–dentin border. In addition, the newly formed hard tissue, such as tertiary dentin or bone-like tissue in the dental pulp, was determined based on nestin-positive or -negative expression around these areas. One week after replantation, the pulpal tissue of the replanted teeth exhibited inflammatory features based on the type and concentration of each experimental solution (Figure 1A,D,G,J,M). Severe inflammatory reactions, including hemorrhagic areas, were observed in the whole dental pulp of replanted teeth treated with CpG-ODN A at 0.8 mM (asterisk in Figure 1G). Re-vascularization did not occur in the coronal pulp of the CpG-ODN B groups at 0.1 mM and 0.8 mM, whereas clear blood lumens were distinguished in the pulpal floor of the DW and CpG-ODN A 0.1 mM groups (double arrows in Figure 1A,D), and the HBSS group (double arrows in Appendix A). Collagen-related blue matrices were faintly observed by AZAN staining in the dental pulp of the DW (arrow in Figure 1B) and HBSS groups (arrow in Appendix A) but barely detected in the CpG-ODN groups (Figure 1E,H,K,N). Immunohistochemistry for nestin revealed the presence of surviving odontoblasts and/or odontoblast-like cells (arrows) aligning under the pulp–dentin border of the replanted teeth, as well as numerous nestin-positive filamentous structures (arrowheads), occupying the pulpal areas where the healing process was occurring (Figure 1C,F,I,L,O and Appendix A). Most of the cells located under the tip of the pulpal horns lacked nestin immunoreactivity in the DW (Figure 1C), HBSS (Appendix A), CpG-ODN A 0.1 mM, and CpG-ODN B 0.1 mM and 0.8 mM groups (Figure 1F,L,O), whereas only scattered nestin-positive cells were detected in the coronal pulp of replanted teeth of the CpG-ODN A 0.8 mM group (Figure 1I). Significant differences in the rate of nestin-positive perimeters over the total perimeter of the dental pulp of replanted teeth were observed between the HBSS and CpG-ODN groups and DW (*p* < 0.05 or *p* = 0.01), particularly those treated with a concentration of 0.8 mM (*p* < 0.01) (Figure 2A).

At week 2, all groups exhibited newly formed hard tissue in the dental pulp, particularly in the root area (asterisks in Figure 3A,E,I,M,Q and Appendix A). Despite the presence of pulpal inflammatory reactions previously observed one week after replantation, the dental pulp of the replanted teeth treated with the CpG-ODN solutions recovered its characteristic features (Figure 3E,I,M,Q), including re-vascularization, which was comparable with those treated with DW (Figure 3A) and HBSS (Appendix A). However, most samples in the DW and CpG-ODN groups and some in the HBSS group showed some root ankylosis at this stage (Figure 3A,E,I,M,Q and Appendix A). AZAN staining revealed that the blue areas related to collagen deposition in the newly formed hard tissue were similar in all experimental groups (Figure 3B,F,J,N,R and Appendix A). A slight positive tendency for the occurrence of newly formed hard tissue in the dental pulp of replanted teeth was observed for the HBSS group and CpG-ODN A and B groups at a concentration of 0.1 mM; however, no significant differences were observed among the groups (Figure 2C). Nestin immunostaining revealed the presence of different healing patterns in the repaired pulpal tissue (Figure 3C,D,G,H,K,L,O,P,S,T and Appendix A). Tertiary dentin, including pulp stones (Figure 3C,D,G,H,O,P) and nestin-negative bone-like tissue (arrows) coexisted in the dental pulp (Figure 3K,L,S,T). Nestin-positive newly differentiated odontoblast-like cells were arranged beneath the tertiary dentin or surrounding the pulp stones. No significant differences were observed in the rate of nestin-positive perimeters relative to the total perimeter of the dental pulp of replanted teeth among the groups; however, the CpG-ODN B 0.8 mM group exhibited the lowest rate compared with the other groups (Figure 2B). In addition, we observed that both CpG-ODN A and B at 0.1 mM concentration, as well as the DW and HBSS groups, tended to favor the deposition of tertiary dentin, which reduced the rate of bone-like tissue formation in the pulpal tissue; however, no significant differences were observed among the groups (Figure 2D). Based on our observations, exposure to low concentrations (0.1 mM) of synthetic CPG-ODN Type A or B solutions appears to cause less damage to pulpal tissue in terms of inflammatory responses and, therefore, a higher occurrence rate of hard tissue formation in the dental pulp. 

### 3.2. Analysis of Cell Proliferation in the Pulpal Tissue by Ki-67 Immunohistochemistry

Cell proliferation is a biological event that precedes the process of cell differentiation in tissues undergoing repair/regeneration processes. Ki-67 immunohistochemistry was performed to evaluate the proliferative activity in the dental pulp of replanted teeth exposed to DW, HBSS, and synthetic CpG-ODN solutions. The analysis of Ki-67-positive cells indicates that the inflammatory reactions associated with the replantation procedures, in particular, the use of CpG-ODN solutions at high concentrations (0.8 mM), affected cell proliferation during the initial stages of the pulpal healing process. A positive tendency for cell proliferation was observed toward the DW and HBSS groups one week after replantation, in which a significant difference (*p* < 0.05) was observed in the coronal and root dental pulp between the DW and CpG-ODN B 0.8 mM groups at this stage (Figure 4A,B). At week 2, the number of proliferative cells in the CpG-ODN groups was restored to the level of the DW and HBSS groups, along with the establishment of healing patterns in the dental pulp. No significant differences were observed among the groups at this stage (Figure 4C,D). Thus, the data suggest that the inhibition of proliferative activity induced by the synthetic CpG-ODN solutions at week 1 ceased until week 2, allowing the differentiation process of hard-tissue-forming cells at week 2.

### 3.3. Assessment of Macrophage Activity in the Dental Pulp by F4/80 and CD206 Immunohistochemistry

Immunohistochemistry for F4/80 and CD206 was performed to evaluate macrophage activity. F4/80 is considered a pan macrophage marker, whereas CD206 is a marker for the M2 subset, macrophages associated with anti-inflammatory activity and tissue repair. One week after intentionally delaying tooth replantation, most samples exhibited a mild to moderate reaction to the F4/80 antibody (Figure 5A,E,I and Appendix A). F4/80 exhibited a dendritic cell-like pattern of expression in the dental pulp. At week 1, F4/80 was expressed by cells located in the central pulp and surrounding cell debris. F4/80-negative areas corresponded with those experiencing inflammatory conditions, where pulpal degeneration had occurred, such as in the pulpal horns. There was no significant difference in the percentage of F4/80-positive areas among all groups in the coronal or root pulp; however, a positive trend was noted toward the CpG-ODN B 0.1 mM group (Appendix A). Interestingly, the presence of M2 macrophages (CD206-positive cells) was scarce at this observation point, particularly in the areas where F4/80-positive cells appeared (Figure 5B,F,J and Appendix A). No significant differences were observed in the percentage of CD-206-positive cells among the groups at week 1, although a positive tendency was observed toward the CpG-ODN A and B 0.1 mM groups in the coronal and root pulp (Figure 6A,B). At week 2, the immunoreaction of F4/80 increased in intensity in the CpG-ODN groups despite the onset of healing in the dental pulp (Figure 5C,G,K and Appendix A), which was primarily located in the subodontoblastic layer, central pulp, and surrounding blood capillaries and pulpal stones (Figure 5G,K), whereas only a few F4/80-positive cells were observed beneath the predentin. There were no significant differences in the F4/80-positive area in the coronal and root dental pulp among the groups at this stage (Appendix A). Regarding the activity of M2 macrophages, a small increase in the rate of CD206-positive cell density was noted at this stage. CD206-positive cells increased in number in the subodontoblast layer and central pulp, even in the areas where F4/80-positive cells were observed (Figure 5D,H,L and Appendix A). Although no significant differences were evident among the groups in the coronal pulp (Figure 6C), a significant difference in CD206-positive cell density was observed in the root pulp between DW and HBSS, and the CpG-ODN A 0.1 mM group, respectively (*p* < 0.05) (Figure 6D). Histological and quantitative data from immunohistochemistry for F4/80 and CD206 suggest that synthetic CPG-ODN solutions appeared to increase the activity of dental pulp macrophages without significant M2 subset commitment in both observation periods because intense expression for F4/80 was still observed at week 2.

### 3.4. TLR9 Immunoexpression in the Afflicted Dental Pulp

Synthetic CpG-ODNs are ligand molecules of TLR9. The ligand–receptor activity induced by the experimental solutions in the different cell populations of the pulpal tissue was evaluated. One week after the operation, TLR9 immunoreactivity was faintly detected in some pulpal cells, the surrounding cell debris matrix, and around the blood capillaries in samples from the DW and HBSS groups (arrows in Figure 7A,D), whereas scattered TLR9-positive cells were observed around the pulpal stones and central pulp at week 2 (Figure 7J and arrows in Figure 7M). Nestin-positive filaments and cells were not correlated with TLR9-positive cells (arrowheads in Figure 7G, and Figure 7P). Some samples that sustained pulpal inflammatory reactions in the CpG-ODN A and B 0.1 mM groups showed clear TLR9 immunoreactivity one week after the operation (Figure 7B,C,E,F). Afflicted cells located under the pulpal horns beneath the pulp–dentin border showed a positive reaction for TLR9 (arrows) and a negative immunoreaction for nestin (Figure 7H, and arrowheads in Figure 7I). At week 2, a less intense TLR9 immunoreactivity was observed in the pulpal cells surrounding the newly formed hard tissue and central pulpal tissue (arrows) of some samples in the CpG-ODN groups (Figure 7K,L,N,O). Some of these cells correlated with nestin-positive cells surrounding the newly formed hard tissue areas (arrowheads in Figure 7Q,R). The quantitative analysis of the percentage of TLR9-cell positive areas in the dental pulp showed no significant differences among groups during both observation periods (Appendix A). Therefore, these novel data suggest that synthetic CpG-ODNs may activate TLR9 in specific cell populations of non-odontoblastic lineage and possibly in dental pulp progenitor cells at week 1. Consequently, some TLR9-positive cells seemed to coincide with the newly differentiated hard-tissue-forming cells surrounding tertiary dentin and/or bone-like tissue areas in the dental pulp at week 2.

## 4. Discussion

To our knowledge, this is the first study describing the effects of synthetic CpG-ODNs on the immunomodulatory response and dental pulp repair using in vivo murine tooth injury models, such as intentionally delayed tooth replantation. The properties, molecular mechanisms, and therapeutic uses of synthetic CpG-ODNs have been extensively reported [1,2,3,4,5,6,7,25,26,27,28,29,30]; however, only a few studies have described their potential use in dentistry [31,32,33]. In the present study, the topical use of synthetic CpG-ODNs following tooth injury elicited an inflammatory reaction and a low rate of cell proliferation in the dental pulp during the first week after treatment. This led to the establishment of two different healing patterns: tertiary dentin and a mixed form of tertiary dentin and bone-like tissue in the dental pulp at week 2 following the operation. However, we did not observe significant differences among the groups because the use of synthetic CpG-ODNs did not show any clear advantages at different stages of the dental pulp repair process, either from a cellular or morphological point of view. Therefore, the concentration of synthetic CpG-ODN solutions and the immersion time may be important factors in the treatment of the afflicted pulp. Our unpublished preliminary data show that treatment with synthetic CpG-ODN A and B at a concentration of 1.57 mM resulted in the strongest pulpal inflammatory reactions among the tested concentrations on week 1 after replantation and the lowest occurrence of hard tissue formation in the dental pulp at week 2, suggesting that exposure to synthetic CpG-ODN solutions at high concentrations worsens the prognosis of the afflicted pulpal tissue. These results are consistent with those of a previous study that analyzed the effects of different concentrations and immersion times of extracted teeth of mice in an experimental triple antibiotic (3Mix) solution before replantation. The immersion of the extracted teeth in a phosphate-buffered saline (PBS) solution, followed by short immersion (5 to 15 min) in the 3Mix solution at a standard (low) concentration, improved both pulpal and periodontal healing [20]. In addition, our preliminary data prove that immersion for 30 min is sufficient to ensure the penetration of CpG-ODNs into the pulpal tissue. Under the same experimental conditions, the extracted teeth of the mice were immersed into a GFP-conjugated CpG-ODN solution and then replanted. One hour after replantation, a GFP signal was detected in the coronal pulp, which remained invariable during the following 24 h (Appendix A). This suggests that CpG-ODNs may have a long-lasting or residual effect on the exposed tissues, which elicits a continued immune response in the pulpal cells. CpG-ODNs with a phosphorothioate backbone, the ones used in this study, are less susceptible to DNase digestion, resulting in longer in vivo half-lives compared with ODNs with a phosphodiester backbone [5]. Based on these structural characteristics, they are optimal immune adjuvants, as they can extend the effect of vaccines [34,35]. Thus, further in vivo studies on the effect of synthetic CpG-ODNs on dental tissue should consider these variables under various conditions, such as shorter immersion times (<30 min), lower concentrations, or the use of transfer solutions (HBSS or DW) before or after immersion, to find the ideal scenario to promote the acceleration of pulpal healing following injury.

This study shows the presence of TLR9-positive cells in the dental pulp of some samples from the HBSS and CpG-ODN groups at weeks 1 or 2 after replantation. To date, TLR9 mRNA expression has been confirmed in human- and murine-differentiated odontoblasts in vitro [11,12,15,16,17,18,19]. Nestin immunostaining confirmed that the TLR9-positive cells observed one week after replantation in the CpG-ODN groups are not of odontoblast lineage but leukocytes or antigen-presenting cells, such as pulpal macrophages or dendritic cells. As CpG-ODNs can directly activate B cells, macrophages, and dendritic cells [1,2,3,4,5,7,36,37], we hypothesize that CpG-ODNs may exert a direct immunostimulatory effect on resident immune cells of the dental pulp following their exposure. Moreover, pulpal progenitors may also be stimulated by direct exposure to CpG-ODNs through the prolonged activation of TLR9. The latter could explain why some TLR9-positive cells were observed around pulpal stones in some samples of the CpG-ODN A 0.1 mM group at week 2 when the healing patterns were already established in the dental pulp of all groups. In the absence of comparable in vivo data, further studies are needed to determine the expression of TLR9 in dental pulp cells under normal and pathological conditions and how CpG-ODN activates dental pulp antigen-presenting cells and/or progenitor cells following dental injuries.

Similarly, this study shows that exposure to synthetic CpG-ODNs tends to stimulate macrophage activity in the coronal and root dental pulp after replantation compared with HBSS and DW. Several recent studies have highlighted the importance of macrophage polarization in the healing of pulpal tissue [38,39,40,41,42]. Macrophages present in different tissues, including the dental pulp, can polarize in response to changes in their environment. For example, M1 macrophages are primarily involved in pro-inflammatory responses by secreting cytokines, such as interleukin 1β (1L-1β), tumor necrosis factor-alpha (TNF-α), IL-6, reactive oxygen species (ROS), and nitric oxide (NO) [41], whereas M2 macrophages can induce an anti-inflammatory response and tissue repair [43,44]. Although we did not analyze the presence of M1 macrophages, we presume that exposure to CpG-ODNs at 0.1 mM may activate the M1 subset because the quantitative analysis of F4/80 immunohistochemistry revealed a slight trend toward CpG-ODNs compared with that in the HBSS and DW groups at week 2. Moreover, the percentage of M2 macrophages in the coronal and root pulp decreased at the same stage, particularly after treatment with CpG-ODNs-A 0.1 mM compared with HBSS and DW. The pro-inflammatory immune response observed in the present study was previously reported by He et al. [15,18,19], albeit using an in vitro approach with murine-differentiated odontoblasts. Based on the premise that CpG-ODN can stimulate inflammation and induce the M1 macrophage phenotype that restricts angiogenesis [45], recent studies have demonstrated that novel delivery approaches of CpG-ODNs can effectively regulate a tumor-associated M2 macrophage transformation into the M1 phenotype [46,47,48], thus facilitating the development of promising cancer therapeutics. In contrast, M2 macrophages were shown to be key to reversing the early stages of pulpal inflammation and achieving wound healing in a rat model for cavity preparation and orthodontic tooth movement [38,40]. Therefore, exposure to synthetic CpG-ODNs may alter the interactions of M1/M2 macrophages with other cell populations, such as resident dendritic cells and dental pulp progenitor cells, which are essential for odontoblast differentiation under pathological conditions [49,50].

Overall, our findings suggest that treatment with low-concentration synthetic CpG-ODN solutions, either Type A or B at 0.1 mM, may evoke a long-lasting macrophage–TLR9-mediated pro-inflammatory, rather than anti-inflammatory, response in the dental pulp. Both types of synthetic CpG-ODNs primarily enhanced the immune responses by activating TLR9 signaling pathways in the dental pulp. Our data show a positive trend toward the use of synthetic CpG-ODN solutions at low concentrations (0.1 mM) to promote healing of the afflicted pulpal tissue better than the 0.8 mM concentration during both observation periods. However, when comparing both CpG-ODN types at 0.8 mM, Type B seems to provide a slightly better situation for pulpal healing than Type A at week 1, especially in the coronal pulp. Based on this observation, it is possible to assume that Type B may have a less adverse effect on the dental pulp than Type A during the initial stages after tooth replantation and, therefore, tends to result in increased hard tissue deposition at week 2. The evidence that Type A CpG-ODN can target pDCs and enhance systemic immunity, while Type B CpG-ODN can induce proinflammatory cytokines from B cells and macrophages, including those present in inflamed pulpal tissue [51,52], supports the tentative assumption that Type B is superior to Type A for the healing of the injured dental pulp, although this claim needs to be further confirmed by additional studies. Despite the disadvantages that may have arisen from our experimental design, such as the prolonged immersion time, we conclude that immersion for 30 min in CpG-ODN solutions may have a detrimental effect on the process of pulpal repair. This might result in the overactivation of the M1 macrophage subset and/or the M2-to-M1 macrophage transformation in the afflicted dental pulp of replanted teeth. Although this study did not provide evidence on this subject, the concerns regarding M1 macrophages must be further analyzed in detail to support our current hypothesis. Shorter immersion times in synthetic CpG-ODNs may effectively activate the M1 macrophage subset, which is needed during the initial inflammatory response for the removal of microbial-derived noxious products or cell debris. The timing in which the activity of M1 macrophages should give way to M2 macrophages to initiate anti-inflammatory activity and the proper balance between M1/M2 polarized macrophages in order to promote tissue repair in the afflicted dental pulp also need to be elucidated. In addition, the mechanisms behind the differentiation of hard-tissue-forming cells and the subsequent deposition of tertiary dentin and/or bone-like tissue, while sustained macrophagic activity continues to occur within the dental pulp, remain to be determined. Further studies analyzing the effects of immunomodulatory agents in vitro and in vivo are needed to develop therapeutic strategies for dental tissue regeneration.

## Figures and Tables

**Figure 1 biomolecules-14-00931-f001:**
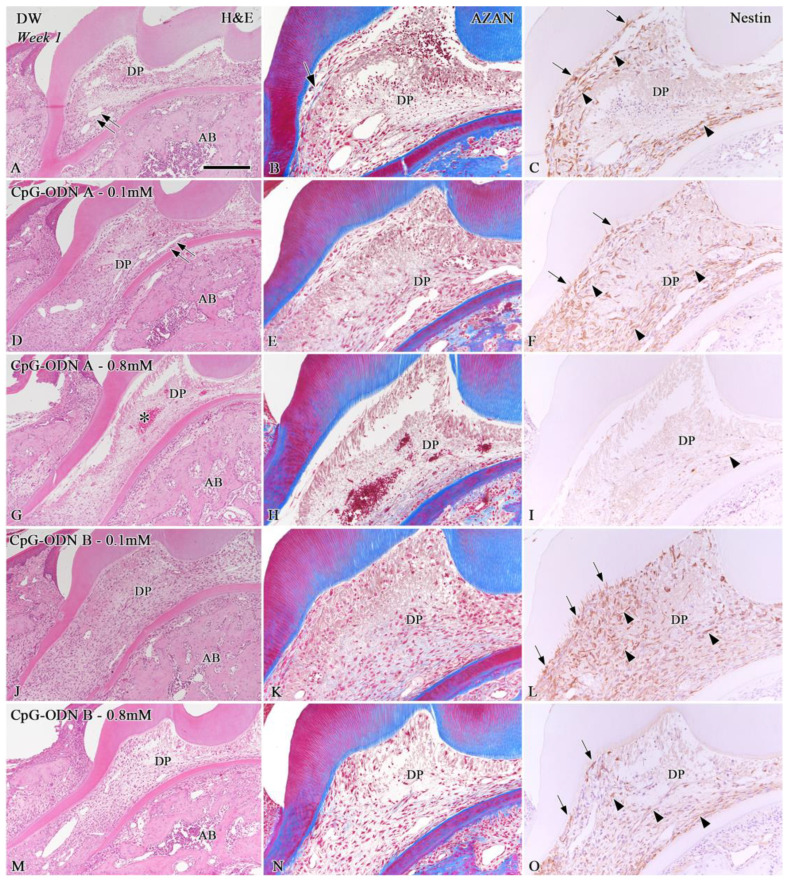
Hematoxylin–eosin (H&E) staining (**A**,**D**,**G**,**J**,**M**), AZAN staining (**B**,**E**,**H**,**K**,**N**), and nestin immunoreactivity (**C**,**F**,**I**,**L**,**O**) in samples of extracted teeth treated with DW (**A**–**C**), CpG-ODN A at 0.1 mM (**D**–**F**), CpG-ODN A at 0.8 mM (**G**–**I**), CpG-ODN B at 0.1 mM (**J**–**L**), and CpG-ODN B at 0.8 mM (**M**–**O**) one week after replantation. (**A**,**D**,**G**,**J**,**M**) The pulpal tissue of all replanted teeth shows inflammatory features. Hemorrhagic areas are observed in CpG-ODN A at 0.8 mM (asterisk in (**G**)). Clear blood lumens are distinguished in the pulpal floor of the DW and CpG-ODN A 0.1 mM groups (double arrows in (**A**,**D**)). (**B**,**E**,**H**,**K**,**N**) Collagen-related matrices are faintly observed in blue color in the DW group (arrow in **B**) but barely in the CpG-ODN groups. (**C**,**F**,**I**,**L**,**O**) Surviving odontoblasts and/or odontoblast-like cells (arrows) align under the pulp–dentin border of replanted teeth, as well as in numerous nestin-positive filamentous structures (arrowheads). AB: Alveolar bone, DP: Dental pulp. Scale bars: 250 µm (**A**,**D**,**G**,**J**,**M**) and 100 µm (**B**,**C**,**E**,**F**,**H**,**I**,**K**,**L**,**N**,**O**).

**Figure 2 biomolecules-14-00931-f002:**
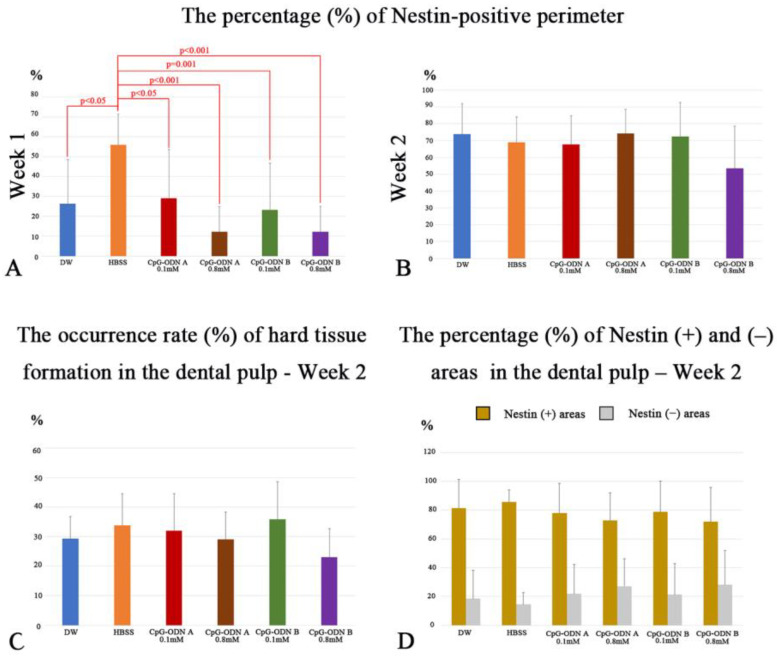
Quantitative analysis of H&E staining and nestin immunohistochemistry. (**A**) The percentage (%) of the nestin-positive perimeter one week after replantation. Significant differences are evident between the HBSS and CpG-ODN groups and DW. (**B**) During week 2, there are no significant differences in the rate of nestin-positive perimeters among the groups. (**C**) The occurrence rate (%) of hard tissue formation in the dental pulp during week 2 shows no significant differences among groups but a slight positive tendency toward the HBSS group and CpG-ODN A and B groups at 0.1 mM. (**D**) There are no significant differences in the percentage (%) of nestin-positive and -negative areas in the dental pulp at week 2 after replantation.

**Figure 3 biomolecules-14-00931-f003:**
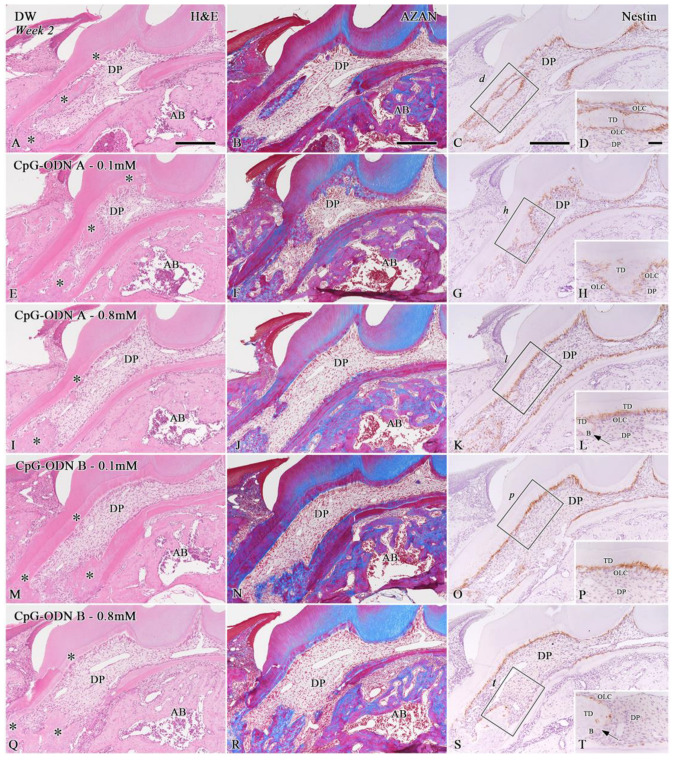
H&E staining (**A**,**E**,**I**,**M**,**Q**), AZAN staining (**B**,**F**,**J**,**N**,**R**), and nestin immunoreactivity (**C**,**D**,**G**,**H**,**K**,**L**,**O**,**P**,**S**,**T**) in samples of extracted teeth treated with DW (**A**–**D**), CpG-ODN A—0.1 mM (**E**–**H**), CpG-ODN A at 0.8 mM (**I**–**L**), CpG-ODN B at 0.1 mM (**M**–**P**), and CpG-ODN B at 0.8 mM (**Q**–**T**) two weeks after replantation. (**A**,**E**,**I**,**M**,**Q**) The dental pulp of the DW and CpG-ODN groups shows re-vascularization and newly formed hard tissue areas (asterisks). (**B**,**F**,**J**,**N**,**R**) The AZAN-stained blue areas are similar for all experimental groups. (**C**,**D**,**G**,**H**,**O**,**P**) Nestin immunostaining reveals the presence of tertiary dentin, including pulp stones. (**K**,**L**,**S**,**T**) Nestin-negative bone-like tissue (arrows) coexists in the dental pulp. (**D**,**H**,**L**,**P**,**T**) Higher magnified views of the boxes in (**C**,**G**,**K**,**O**,**S**), respectively. AB: Alveolar bone, B: Bone-like tissue, DP: Dental pulp, OLC: Odontoblast-like cells, TD: Tertiary dentin. Scale bars: 250 µm (**A**–**C**,**E**–**G**,**I**–**K**,**M**–**O**,**Q**–**S**) and 50 µm (**D**,**H**,**L**,**P**,**T**).

**Figure 4 biomolecules-14-00931-f004:**
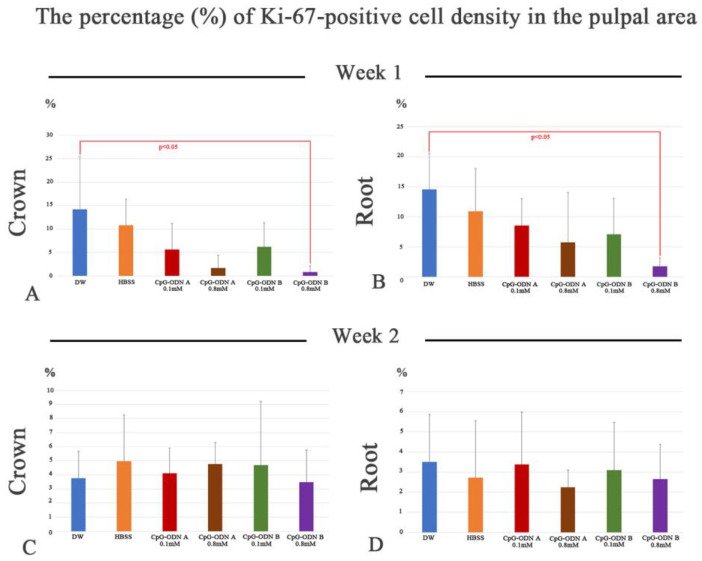
The percentage (%) of Ki-67-positive cell density in the coronal (**A**,**C**) and pulpal (**B**,**D**) root area at weeks 1 (**A**,**B**) and 2 (**C**,**D**) after replantation in the DW, HBSS, and CpG-ODN A and B 0.1 mM groups. (**A**,**B**) The number of Ki-67-positive cells decreases particularly in the CpG-ODN groups one week after replantation. A significant difference is noted between DW and CpG-ODN B at 0.8 mM in the coronal and root pulp. (**C**,**D**) At week 2, the number of proliferating cells in the CpG-ODN groups reaches the level of the DW and HBSS groups. No significant differences are observed among the groups.

**Figure 5 biomolecules-14-00931-f005:**
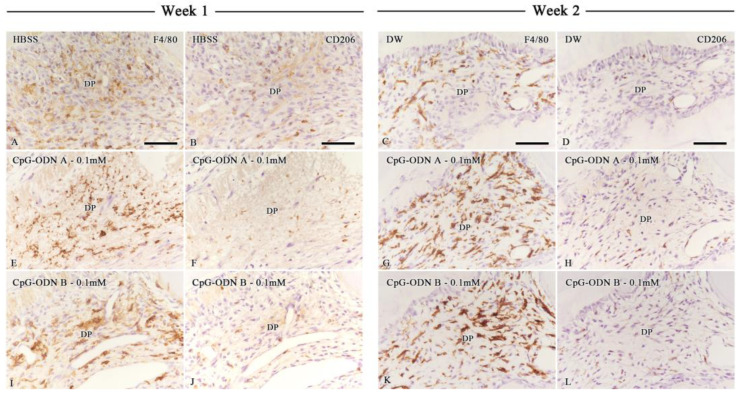
Immunohistochemical evaluation of macrophage activity in dental pulp. F4/80 (**A**,**C**,**E**,**G**,**I**,**K**) and CD206 (**B**,**D**,**F**,**H**,**J**,**L**) immunohistochemistry in the HBSS (**A**,**B**) and DW (**C**,**D**) groups and the CpG-ODN A (**E**–**H**) and CpG-ODN B (**I**–**L**) 0.1 mM groups at weeks 1 (**A**,**B**,**E**,**F**,**I**,**J**) and 2 (**C**,**D**,**G**,**H**,**K**,**L**) following replantation. (**A**,**E**,**I**) Most samples exhibit a mild to moderate reaction to the F4/80 antibody, with negative areas corresponding to those under inflammatory conditions one week after replantation. (**C**,**G**,**K**) At week 2, F4/80 immunoreactivity increases in intensity in the pulpal tissue, particularly in the subodontoblast layer, central pulp, and areas surrounding the blood capillaries and pulpal stones in the CpG-ODN A and B 0.1 mM groups. (**B**,**F**,**J**) CD206-positive cells are scarce at week 1, particularly in the areas where F4/80-positive cells are observed. (**D**,**H**,**L**) At week 2, CD206-positive cells increase in number in the subodontoblast layer and central pulp, even in the areas where F4/80-positive cells are observed. DP: Dental pulp. Scale bar: 50 µm (**A**–**L**).

**Figure 6 biomolecules-14-00931-f006:**
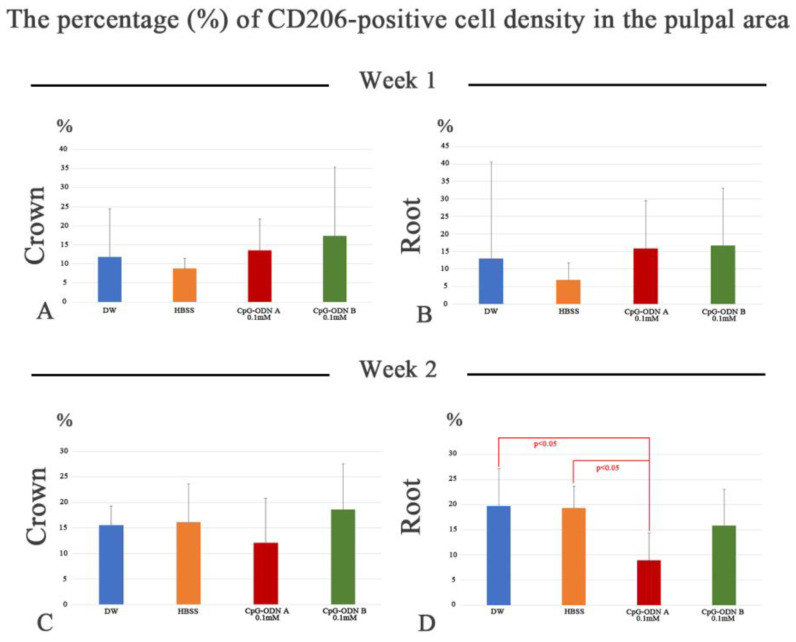
The percentage (%) of CD206-positive cells in the coronal (**A**,**C**) and pulpal (**B**,**D**) root area at weeks 1 (**A**,**B**) and 2 (**C**,**D**) after replantation in the DW and HBSS groups and the CpG-ODN A and B 0.1 mM groups. (**A**,**B**) There are no significant differences among the groups at week 1, although a positive tendency is evident toward the CpG-ODN A and B 0.1 mM groups in the coronal and root pulp. (**C**,**D**) At week 2, there is no significant difference among the groups in the coronal pulp, whereas a significant difference is found in the root pulp between the DW and HBSS groups and CpG-ODN A 0.1 mM group, respectively.

**Figure 7 biomolecules-14-00931-f007:**
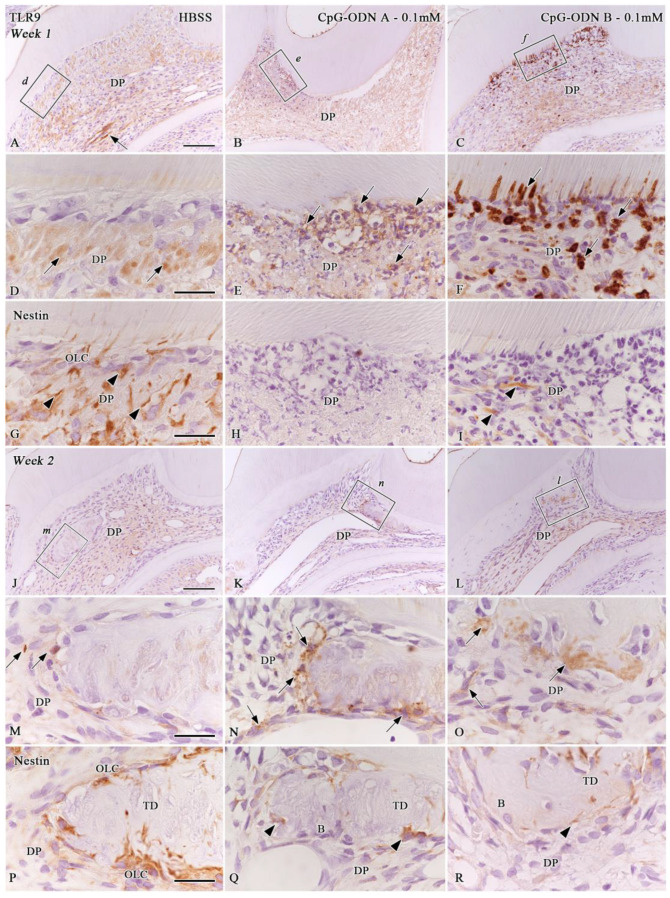
The immunoexpression of TLR9 (**A**–**F**,**J**–**O**) and nestin immunohistochemistry (**G**–**I**,**P**–**R**) in the dental pulp of replanted teeth treated with HBSS (**A**,**D**,**G**,**J**,**M**,**P**) and CpG-ODN A (**B**,**E**,**H**,**K**,**N**,**Q**) and B (**C**,**F**,**I**,**L**,**O**,**R**) at 0.1 mM at weeks 1 (**A**–**I**) and 2 (**J**–**R**), following replantation. (Arrows in (**A**,**D**)) TLR9 immunoreactivity is faintly detected in samples from the HBSS group at week 1. (**B**,**C**,**E**,**F**) TLR9 immunoreactivity is observed in some samples that sustained pulpal inflammatory reactions in the CpG-ODN groups at this stage (arrows in (**E**,**F**)). (**G**–**I**) Nestin-positive filaments (arrowheads) and cells do not correlate with TLR9-positive cells. (**J**,**M**) Scattered TLR9-positive cells (arrows in (**M**)) are observed around pulpal stones in the HBSS group at week 2. (**K**,**L**,**N**,**O**) TLR9 immunoreactivity is observed in cells surrounding pulpal stones of some samples in the CpG-ODN groups (arrows in (**N**,**O**)). (**P**–**R**) TLR9-positive cells partially correlate with nestin-positive cells at this stage (arrowheads in (**Q**,**R**)). (**D**–**I**,**M**–**O**) Magnified views of the boxed areas in and (**A**,**J**–**L**), respectively. B: Bone-like tissue, DP: Dental pulp, OLC: Odontoblast-like cells, TD: Tertiary dentin. Scale bars: 100 µm (**A**–**C**,**J**–**L**) and 25 µm (**D**–**I**,**M**–**R**).

## Data Availability

The original data presented in the study are included in the article/Appendix A. Further inquiries can be directed to the corresponding author.

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
