# Peer review of "Effects of Synthetic Toll-Like Receptor 9 Ligand Molecules on Pulpal Immunomodulatory Response and Repair after Injuries"

_biomolecules, 2024, doi:10.3390/biom14080931_

Round 1

Reviewer 1 Report

Comments and Suggestions for Authors

In the manuscript entitled “Effects of synthetic Toll-like receptor 9-ligand molecules on 2 pulpal immunomodulatory response and repair after injuries”. The authors assessed the effects of CpG-15 ODNs on pulpal immunomodulatory response and repair after injury. Although the study is interesting, it is still preliminary and need further work, especially in the context of immunomodulatory response. Here are my concerns

1.   Did the authors test and optimize different concentrations of CpG-ODN A and B with different exposure time.

2.   To evaluate the effect on the collagen fibers, only AZAN staining was used. Protein expression specific for collagens is needed for confirmation.

3.   Regarding the immunomodulatory response, it is important to evaluate other immune cells such as B and T lymphocytes, NK cells.

4.   In figure 1, low concentration of CpG-ODN A showed higher Nestin expression compared to 0.8mM. However, in B concentration showed high expression. What is the reason?

5.   Also, at week 1 there is an increase in the expression of Nestin but at week 2, no differences were observed among the groups means that CpG-ODN has short action effect?

6.   M2 macrophages, a subtype of macrophages that play a critical role in tissue repair and anti-inflammatory. In the study, no difference was observed in the percentage of CD-206-positive cells among the groups. What about other macrophage markers.

Comments on the Quality of English Language

English is fine

Author Response

To reviewer #1

In the manuscript entitled “Effects of synthetic Toll-like receptor 9-ligand molecules on pulpal immunomodulatory response and repair after injuries”. The authors assessed the effects of CpG-ODNs on pulpal immunomodulatory response and repair after injury. Although the study is interesting, it is still preliminary and need further work, especially in the context of immunomodulatory response. Here are my concerns

Thank you for your kind suggestions. The points of improvement are as follows.

  1. Did the authors test and optimize different concentrations of CpG-ODN A and B with different exposure time.

We appreciate the reviewer’s evaluation. Yes, in this project we tested three different concentrations of CpG-ODN A and B: 0.1mM, 0.8mM and 1.57mM, resulting in the latter being the most harmful for the healing of the pulpal tissue, and therefore, not included in this manuscript. Regarding the exposure time, we tested both immediate (1 min.), and delayed exposure times (30 and 60 min.) to CpG-ODN A and B solutions. The results obtained from our injury model after 1 min. of exposure to the experimental solutions did not differ from those obtained in DW and HBSS groups under the same conditions. In addition, we did not find major differences between the 60- and 30-minutes exposure time, wherefore the exposure time was determined at 30 minutes.

  1. To evaluate the effect on the collagen fibers, only AZAN staining was used. Protein expression specific for collagens is needed for confirmation.

We understand the reviewer`s notion. We used AZAN staining to roughly visualize the deposition of collagen in the pulpal tissue as another general indicator of the progression of the healing process, since AZAN staining cannot identify all types of collagen fibers. A previous study by our group (Saito K, Nakatomi M, Ida-Yonemochi H, Ohshima H. Osteopontin Is Essential for Type I Collagen Secretion in Reparative Dentin. J Dent Res. 2016 Aug;95(9):1034-41.) showed that the immunoexpression of Collagen type 1 in the dental pulp was intensely seen in the odontoblast layer and predentin, and did not correlate with the AZAN-postive areas. We agree with the reviewer`s notion on this matter and consequently we modified the sentences related to the collagen deposition evidenced by AZAN staining in the results sections (highlighted in yellow color).

  1. Regarding the immunomodulatory response, it is important to evaluate other immune cells such as B and T lymphocytes, NK cells.

We totally agree with the reviewer’s notion. Specific immunohistochemical markers are important for the evaluation of B lymphocytes, T lymphocytes, and NK cells. Currently, we do not have any suitable antibodies for lymphocytes or NK cells that work in our mouse paraffin sections, but we will definitely consider this issue in our future studies. In general, lymphocytes may be evaluated in the H&E stained sections, but it is not possible to distinguish B from T types. Following reviewer`s suggestion, we would like to assess the presence of lymphocytes or NK cells in the pulpal tissue in the future work.

  1. In figure 1, low concentration of CpG-ODN A showed higher Nestin expression compared to 0.8mM. However, in B concentration showed high expression. What is the reason?

We understand the reviewer`s notion. In our study, the use of Nestin immunostaining allowed clarification of the pulp healing or degeneration process after CpG-ODNs exposure and tooth replantation. At week 1 (Figure 1), surviving and/or newly-differentiated odontoblasts aligning under the predentin showed the Nestin positive expression, whereas other pulpal cells were also positive, thus increasing its expression throughout the pulpal tissue. One week after replantation, our data shows that low concentrations (0.1 mM) of CpG-ODN type A or B promoted the healing process of the afflicted tissue better than the 0.8 mM concentration. However, when comparing both CpG-ODN types at 0.8 mM, the type B seems to provide a slightly better situation for the pulpal healing that type A, especially in the coronal pulp. This observation led us to assume that Type B may have a less harmful effect in the dental pulp than Type A during the initial stages after tooth replantation, and this could be the reason of the differences pointed in the reviewer`s comment.

  1. Also, at week 1 there is an increase in the expression of Nestin but at week 2, no differences were observed among the groups means that CpG-ODN has short action effect?

We agree with the reviewer`s notion regarding the short action effect of CpG-ODNs. The graphs in Figure 2A and B focus on the quantification of Nestin immunoexpression at the pulp-dentin border. Significant differences were observed between HBSS and all other groups at week 1, but not at week 2. Although the tested conditions were not suitable for observing significant differences, we hypothesize that exposure to CpG-ODNs evoked the temporal or short-term activation of pulp macrophages during the initial stages after replantation (probably the M1 subset), which may have triggered a further inflammatory state, not necessarily linked to the presence of CpG-ODNs in the pulpal tissue, thus inhibiting the healing of the dental pulp in replanted teeth. Future studies by our group will analyze how CpG-ODNs activate macrophage subsets in a time- and concentration-dependent manner in the regenerating dental pulp, especially during the initial stages after replantation.

  1. M2 macrophages, a subtype of macrophages that play a critical role in tissue repair and anti-inflammatory. In the study, no difference was observed in the percentage of CD-206-positive cells among the groups. What about other macrophage markers.

We understand the reviewer`s notion. In our study beside CD206, we provided evidence of the total macrophage activity by analyzing F4/80 immunohistochemistry. The percentage of F4/80 positive areas in the coronal and root pulp of DW, HBSS, and CpG-ODN type A and B at 0.1mM was evaluated at weeks 1 and 2 after tooth replantation, as shown in Supplementary Fig. 4. In both observation periods, the highest total macrophage activity was induced by CpG-ODN type B in the coronal and root pulp. We assume that this percentage was likely mainly influenced by the activity of the M1 macrophage subset, since the M2 subset showed low values, especially at week 1 after replantation. At present, there are no other good markers working in our mouse paraffin sections to assess macrophage activity, however, the concerns of M1 macrophage subset activity and its relation with other pulpal cell populations is the part of our future work.

         Thank you again for your criticism.

Reviewer 2 Report

Comments and Suggestions for Authors

In this study, Angela et al. examined the effects of CpG-ODNs on pulpal immunomodulatory response and repair after injury in mice. Molar teeth were treated with CpG-ODN solutions and replanted. Results showed inflammatory reactions in all groups one week post-operation, with revascularization and new hard tissue formation observed at two weeks. Low concentrations of CpG-ODNs promoted a macrophage-TLR9-mediated pro-inflammatory response, suggesting their potential in modulating dental pulp repair and hard tissue formation. However, a few important questions must be addressed before considering publication.

1. The novelty of this study does not fit general interests. The authors showed that 30min treatment does not provide expected results or even a reversed result but lack follow up experiments or mechanistic studies. They should examine how CpG-DNs behave as pro-inflammatory factors in this case.

2. It is unclear to this reviewer that why the authors did a treatment with HBSS. The CpG-DNs are dissolved in water and thus water should be the correct control to be shown in figure 1.

3. The authors claimed that “This results in the over-activation of the M1 macrophage subset and/or the M2 to M1 macrophage transformation in the afflicted dental pulp of replanted teeth.” (Line 435) However, they did not evident the presence of M1 macrophage. They should perform immunostaining of M1 macrophage marker to support their claim. This will be an important data for their claim.

4. In the results section, the authors did a good job describing their data. However, there is no conclusion for these results. The authors should provide a clear statement about their conclusion for each result. The audiences are interested in what their data means, rather than just seeing the data. Moreover, they should provide rationale for each experiment. This will make the story flows better and help the audience to understand their work.

5. The author did not provide enough information about the sample size. Line 116, “Samples were collected from groups of 2–5 animals on weeks 1 and 2 following the 116 operation.” What is the exact number of animals for each group? 2 animals are not acceptable considering the variations between animals so I would encourage the authors to increase sample size for conditions with sample size less than 3. Dot plot will be helpful.

6. The authors should use arrows to indicate the structure they mentioned about in the figures. For example, in figure 1, blood lumen, Nestin-positive filamentous structures, and other structures that are mentioned in the manuscript should be clearly indicated.

7. It seems that class A and class B CpG-DNs have very different effects. The authors should provide information about the differences between these two CpG-DNs and discuss why the effects could be different.

Author Response

To reviewer #2

In this study, Angela et al. examined the effects of CpG-ODNs on pulpal immunomodulatory response and repair after injury in mice. Molar teeth were treated with CpG-ODN solutions and replanted. Results showed inflammatory reactions in all groups one week post-operation, with revascularization and new hard tissue formation observed at two weeks. Low concentrations of CpG-ODNs promoted a macrophage-TLR9-mediated pro-inflammatory response, suggesting their potential in modulating dental pulp repair and hard tissue formation. However, a few important questions must be addressed before considering publication.

Thank you for your kind suggestions. The points of improvement are as follows.

  1. The novelty of this study does not fit general interests. The authors showed that 30min treatment does not provide expected results or even a reversed result but lack follow up experiments or mechanistic studies. They should examine how CpG-ODNs behave as pro-inflammatory factors in this case.

We appreciate the reviewer’s evaluation. As mentioned in the discussion section, the results of our study did not show clear significant differences among groups, probably due to the concerns of long immersion times or the concentrations of CpG-ODN solutions. Regarding the mechanisms elicited by CpG-ODNs treatments at the cellular level, previous studies conducted by He W. et al. published between 2010–2014, showed that different types of synthetic CpG-ODNs (ODN 1668, ODN 1720, and ODN 2006) induced pro-inflammatory cytokines, such as IL-6, IL-8, TNF-α, and MMP13 in the mouse odontoblast-like cell line MDPC-23. Furthermore, this pro-inflammatory activity was shown to be mediated by the TLR9, MyD88, NF-κB, and ERK pathways. Considering the existence of in vitro data, this study focused on the evaluation of the pulpal responses in vivo. Although follow-up observations are recommended to evaluate the long-term prognosis of replanted teeth, our previous studies (i.e., Quispe-Salcedo A., Ida-Yonemochi H., Ohshima H. The effects of enzymatically synthesized glycogen on the pulpal healing process of extracted teeth following intentionally delayed replantation in mice. J. Oral Biosci. 2015;57:124–130) demonstrated that the healing patterns (e.g. tertiary dentin,  bone-like tissue, or a combination of both) observed in the dental pulp two weeks after replantation do not change at week 3 onwards when newly-formed hard tissue occupies the majority of the pulpal space. However, we agree with the reviewer`s notion that pro-inflammatory activity should be evaluated in vivo. We will consider these valuable suggestions for future studies that can specifically evaluate the inflammatory markers after exposure to CpG-ODNs solutions.

  1. It is unclear to this reviewer that why the authors did a treatment with HBSS. The CpG-ODNs are dissolved in water and thus water should be the correct control to be shown in figure 1.

We understand the reviewer`s notion. In our study we considered Hanks Balanced Salt Solution (HBSS) as a control solution together with DW because HBSS is commonly used by clinicians, and indicated as an ideal storage media for avulsed teeth because of its ability of preserving the viability of periodontal ligament cells (Poi WR, Sonoda CK, Martins CM, Melo ME, Pellizzer EP, de Mendoncja MR, Panzarini SR. Storage media for avulsed teeth: a literature review. Braz Dent J. 2013 Sep-Oct;24(5):437–445). In Fig. 2A, HBSS showed the highest percentage of Nestin-positive perimeter at week 1, and therefore we chose it over DW to exemplify the contrast in the pulpal inflammatory conditions with other CpG-ODN groups. Nevertheless, we have modified the panel of Fig. 1 showing the results of DW group instead of HBSS according to the reviewer`s suggestion. The images from HBSS group at week 1 are now Supplementary Fig. 1.

  1. The authors claimed that “This results in the over-activation of the M1 macrophage subset and/or the M2 to M1 macrophage transformation in the afflicted dental pulp of replanted teeth.” (Line 435) However, they did not evident the presence of M1 macrophage. They should perform immunostaining of M1 macrophage marker to support their claim. This will be an important data for their claim.

We totally agree the reviewer`s notion. The concerns regarding the M1 macrophage subset are very important to support our claim. However, at present we do not have a good M1 macrophage marker suitable for paraffin sections in mice, but we plan to address this issue in our future studies. Nevertheless, we have modified the discussion section on Line 487 to read as follows: “This might result in the over-activation of the M1 macrophage subset and/or the M2 to M1 macrophage transformation in the afflicted dental pulp of replanted teeth. Although this study did not provide evidence on this subject, the concerns regarding M1 macrophages must be analyzed further in detail to support our current hypothesis.” according to the reviewer`s suggestion. The additional information is highlighted in yellow color.  

  1. In the results section, the authors did a good job describing their data. However, there is no conclusion for these results. The authors should provide a clear statement about their conclusion for each result. The audiences are interested in what their data means, rather than just seeing the data. Moreover, they should provide rationale for each experiment. This will make the story flows better and help the audience to understand their work.

We totally agree the reviewer`s notion. We have modified the results section adding a statement about the conclusion at the end of each subsection, and the rationale for each experiment at the beginning of the paragraphs as suggested by the reviewer. The additional information is highlighted in yellow color.

  1. The author did not provide enough information about the sample size. Line 116, “Samples were collected from groups of 2–5 animals on weeks 1 and 2 following the operation.” What is the exact number of animals for each group? 2 animals are not acceptable considering the variations between animals so I would encourage the authors to increase sample size for conditions with sample size less than 3. Dot plot will be helpful.

We understand the reviewer`s notion. We apologize for the confusing statement regarding sample size. In this study, many experimental trials were performed using the same animal injury model, so there were cases where there were two mice per experimental group in a given trial. However, as we did not change the experimental design or the conditions of each trial, we combined the total number of animals for histological and quantitative assessments. In addition, we used the upper left and right molars as study units, collecting 2 samples per animal (i.e., two mice = four samples, four mice = eight samples). We modified the Supplementary Table 1 to add information about sample size as suggested by the reviewer, and reported the number of samples in each group at each observation time point. In addition, we have modified the description on Line 114 and now it read as follows: “Briefly, the upper left and right first molars of each animal (one mouse, n = two teeth)…”, and on Line 124 that now reads as follows: “Samples were collected from groups of 5-11 animals in total, on weeks 1 and 2 following the operation.” as suggested by the reviewer. These changes can be observed highlighted in yellow color.

  1. The authors should use arrows to indicate the structure they mentioned about in the figures. For example, in figure 1, blood lumen, Nestin-positive filamentous structures, and other structures that are mentioned in the manuscript should be clearly indicated.

We totally agree the reviewer`s notion. Arrows, double arrows, and arrowheads were included in the figure plates and described in the figure legend according to the reviewer’s suggestion. The additional information is highlighted in yellow color in the texts of the results section and figure legends.

  1. It seems that class A and class B CpG-DNs have very different effects. The authors should provide information about the differences between these two CpG-DNs and discuss why the effects could be different.

We totally agree the reviewer`s notion. CpG-ODNs are mainly categorized into two groups. The following information was added to the introduction on Line 45 to read as follows: “As to the structural and functional differences between both types, CpG-ODNs Type A contain a central palindromic CpG motif and form higher-order structures. They are primarily used for their strong interferon response, targeting plasmacytoid dendritic cells (pDCs), and eliciting systemic immunity. CpG-ODNs Type A also induces small amounts of inflammatory cytokines such as IL-6. In contrast, CpG-ODNs Type B form linear structures and induce IL-6 secretion from TLR9 expressing cells, mainly B and macrophage cells in mouse. This results in the activation of B cells, promoting their proliferation and differentiation [1,2].” In addition, we enriched the discussion adding the following paragraph on Line 470 to read as follows: “Apparently, both types of synthetic CpG-ODNs primarily enhanced the immune responses by activating TLR9 signaling pathways in dental pulp. Our data showed a positive trend towards the use of synthetic CpG-ODN solutions at low concentration (0.1 mM) to promote healing of the afflicted pulpal tissue better than the 0.8 mM concentration during both observation periods. However, when comparing both CpG-ODN types at 0.8 mM, Type B seems to provide a slightly better situation for the pulpal healing than Type A at week 1, especially in the coronal pulp. Based on this observation, it is possible to assume that Type B may have a less adverse effect on the dental pulp than Type A during the initial stages after tooth replantation, and therefore tends to result in increased hard tissue deposition at week 2. The evidence that CpG-ODN Type A can target pDCs and enhance systemic immunity, while CpG-ODN Type B can induce proinflammatory cytokines from B cells and macrophages, including those present in inflamed pulpal tissue [51,52], supports the tentative assumption that Type B is superior to Type A for the healing of the injured dental pulp, although this claim needs to be further confirmed by additional studies.” as suggested by the reviewer. The additional information is highlighted in yellow color.

Thank you again for your criticism.

Round 2

Reviewer 1 Report

Comments and Suggestions for Authors

The authors addressed most of the comments, and they are planning to investigate further issues.  

Comments on the Quality of English Language

fine to me

Reviewer 2 Report

Comments and Suggestions for Authors

The authors fully addressed all my concerns. I believe this work now merits a publication at this journal. Congrats!